# Effects of Exercise on Cardiac Function Outcomes in Women Receiving Anthracycline or Trastuzumab Treatment for Breast Cancer: A Systematic Review and Meta-Analysis

Pedro Antunes [1,2,*], Dulce Esteves [1], Célia Nunes [3], Anabela Amarelo [2,4], José Fonseca-Moutinho [5], Vera Afreixo [6], Henrique Costa [7], Alberto Alves [2,8] and Ana Joaquim [2,4]

1 Research Center in Sport Sciences, Health and Human Development, CIDESD, University of Beira Interior, 6201-001 Covilhã, Portugal; desteves@ubi.pt
2 Associação de Investigação de Cuidados de Suporte em Oncologia (AICSO), 4410-406 Vila Nova de Gaia, Portugal; anabelaamarelo@gmail.com (A.A.); ajalves@ismai.pt (A.A.); anaisabeljoaquim@gmail.com (A.J.)
3 Department of Mathematics and Center of Mathematics and Applications, University of Beira Interior, 6201-001 Covilhã, Portugal; celian@ubi.pt
4 Medical Oncology, Centro Hospitalar Vila Nova de Gaia/Espinho, 4400-129 Vila Nova de Gaia, Portugal
5 Faculty of Health Sciences, University of Beira Interior, 6201-001 Covilhã, Portugal; jafm@ubi.pt
6 CIDMA—Center for Research and Development in Mathematics and Applications, University of Aveiro, 3810-193 Aveiro, Portugal; vera@ua.pt
7 Psychiatry and Mental Health Department, Centro Hospitalar de Setúbal, 2190-446 Setúbal, Portugal; hvscosta@gmail.com
8 Research Center in Sports Sciences, Health Sciences and Human Development, CIDESD, University Institute of Maia—ISMAI, 4475-690 Maia, Portugal
* Correspondence: pantunes_14@hotmail.com

**Abstract:** Background: we conducted a systematic review and meta-analysis of randomized controlled trials (RCTs) to evaluate the efficacy of exercise training on cardiac function and circulating biomarkers outcomes among women with breast cancer (BC) receiving anthracycline or trastuzumab-containing therapy. Methods: PubMed, EMBASE, Cochrane Library, Web of Science and Scopus were searched. The primary outcome was change on left ventricular ejection fraction (LVEF). Secondary outcomes included diastolic function, strain imaging and circulating biomarkers. Results: Four RCTs were included, of those three were conducted during anthracycline and one during trastuzumab, involving 161 patients. All trials provided absolute change in LVEF (%) after a short to medium-term of treatment exposure (≤6 months). Pooled data revealed no differences in LVEF in the exercise group versus control [mean difference (MD): 2.07%; 95% CI: −0.17 to 4.34]. Similar results were observed by pooling data from the three RCTs conducted during anthracycline. Data from trials that implemented interventions with ≥36 exercise sessions (*n* = 3) showed a significant effect in preventing LVEF decline favoring the exercise (MD: 3.25%; 95% CI: 1.20 to 5.31). No significant changes were observed on secondary outcomes. Conclusions: exercise appears to have a beneficial effect in mitigating LVEF decline and this effect was significant for interventions with ≥36 exercise sessions.

**Keywords:** breast cancer; cardiac function; cardiotoxicity; exercise

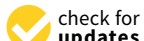



## 1. Introduction

Breast cancer (BC) is the most common malignant disease among women, accounting for more than two million new cases diagnosed worldwide in 2018 [1]. Since 2004, we are noticing a slight increase in BC incidence rate, by 0.3% per year, a trend that is expected to continue to rise over the coming decades [2]. Fortunately, this trend has been followed by the progress in cancer management, including primary prevention, screening, and curative-intent therapies, leading to a notable improvement in specific survival. Globally, the five-years survival rate of women with BC is 91% [2]. As a result of this growing

population of survivors, patients living with and beyond BC face a multiple short and long-term treatment-related side effects [3,4].

Cardiotoxicity has been raised as a major concern in clinical practice, that may restrict or delay treatment options [5], contributing to long-term cardiovascular morbidity and mortality among women with history of BC [6,7]. Despite playing a crucial role for BC treatment, the exposure to drugs such as anthracyclines or to the anti-HER2 newer molecular targeted therapies such as trastuzumab are clinically associated with cardiotoxicity [8], usually manifested by, but not limited to, left ventricular dysfunction to overt heart failure [5,9]. Currently, recommendations for monitoring treatment-related cardiotoxicity entail clinical examination and serial echocardiographic assessments [10]. Resting left ventricular ejection fraction (LVEF) is the standard marker for reporting anticancer therapy-related cardiotoxicity [10]. Cardiotoxicity is defined by a decrease in left ventricular ejection fraction (LVEF) of >10% to a value below the lower limit of normal [9,10]. However, the determination of LVEF has substantial technical limitations and is considered a poor sensitivity parameter to detect cardiotoxicity at an early stage [10]. As such, an integrated approach that includes the assessment of other echocardiographic outcomes (diastolic function and strain imaging) and circulating biomarkers has been recommended [10,11]. Besides the importance of screening and detection, the implementation of preventive strategies to optimize cardiovascular health of BC patients treated with cardiotoxic therapies is also an important component to be considered.

Exercise training is a core component of cardiac rehabilitation and has a major role in primary and secondary prevention of cardiovascular disease [12]. In oncology, exercise has been widely accepted to be a safe and tolerable approach, during or after treatment, playing a beneficial effect on a wide range of physiological and subject-reported outcomes [13–15]. Furthermore, exercise has been associated with a substantial decrease in the risk of cardiovascular disease [16,17] and, thus, there has been an increasing interest in exploring the role of exercise for mitigating cancer treatment-related cardiotoxicity.

This systematic review aims to analyze the effects of exercise, performed during anthracycline or trastuzumab-containing therapy (TC-T), on conventional cardiac function outcomes and circulating biomarkers among women with BC.

## 2. Materials and Methods

### 2.1. Protocol and Registry

This systematic review is registered on PROSPERO (registry number: CRD42017054833) and was conducted according to a previously published protocol [18], which addressed the PRISMA recommendations [19].

### 2.2. Data Sources and Search Terms

Our search strategy was conducted up to 18 May 2020 in: MEDLINE (via PubMed), EMBASE, the Cochrane Central Register of Controlled Trials (CENTRAL), Web of Science and Scopus. No restriction were applied. The references of the included manuscripts and relevant reviews were checked. We also searched three trial register platforms: ClinicalTrials.gov (https://clinicaltrials.gov/)(accessed on 24 May 2020), World Health Organization trials portal (https://www.who.int/clinical-trials-registry-platform)(accessed on 24 May 2020), International Clinical Trials Registry (https://www.isrctn.com/)(accessed on 24 May 2020) to identify ongoing trials. The search strategy were defined by a consensus among the authors of this study and included keywords related to: (1) condition (i.e., breast cancer); (2) treatment (i.e., anthracycline, trastuzumab); (3) exercise (i.e., aerobic exercise, resistance exercise); and (4) study design (i.e., RCT). The text terms and indexing terms were modified according to the searched database. Details of search strategy used in MEDLINE are outlined in supplementary material in Table S1.

### 2.3. Eligibility Criteria

- Study type: published randomized controlled trials (RCTs) in English, French, Germany, Portuguese or Spanish languages.
- Participants: trials involving adult women with BC undergoing neoadjuvant or adjuvant anthracycline-containing therapy (AC-T) or TC-T.
- Intervention: Trials delivering an exercise intervention involving aerobic exercise (any exercise form that uses large muscle groups which predominately stresses the cardiovascular system) and/or resistance exercise (any exercise form that requires a muscle or a muscle group to work against external resistance which predominately stresses the musculoskeletal system), comparing to a non-exercise group. To be eligible, the exercise intervention had to be performed during AC-T or TC-T. Trials were not considered if Yoga, Tai Chi Chuan, Qigong or Pilates was defined as the exercise modality.
- Outcomes: Resting LVEF was defined as primary outcome. Secondary outcomes included diastolic function (E/A' ratio, isovolumetric relaxation time, E/E' lateral, and E/E' septal), strain imaging and circulating biomarkers (troponin I or T; high-sensitivity troponin I or T; brain natriuretic peptide; amino terminal of B-type natriuretic peptide or n-terminal pro-b-type natriuretic peptide).

### 2.4. Data Management and Analysis

The search results were managed using the EndNote software X9 version (Clarivate Analytics, Philadelphia, PA, USA). Two authors (P.A., D.E.) independently reviewed the titles and abstracts of the retrieved results and selected the studies that appeared to meet the predefined population-intervention-control outcome (PICO) criteria. Then, the same two authors performed a non-blind full-text screening to take a final decision. Reasons for exclusion were recorded. Disagreements were resolved by discussion or by judgement of a third author (A.J.). Three authors (P.A., C.N., V.A.) independently extracted data from the included RCTs. Two authors (An.A., H.C.) extracted data from the identified ongoing trials. In both processes, it was used a predesigned data collection form (details in Supplementary Material, File 1).

### 2.5. Risk of Bias Assessment

Two authors (An.A., A.J.) independently assessed risk of bias using the Cochrane collaboration's risk-of-bias tool [20]. Random sequence generation, allocation concealment, blinding of participants and personnel, incomplete outcome data, selective reporting of outcomes, other possible sources of bias and, additional the adherence to exercise intervention were graded as high ('+'), low ('−') or unclear ('?') quality.

### 2.6. Statistical Analysis

We used the mean difference (MD) as the effect size measure to assess the effect of exercise on several quantitative outcomes. We calculated the MD by subtracting the mean change in the exercise group from the mean change in the control group. Homogeneity among studies was computed using the Cochran's Q statistic and $I^2$ statistic. The $I^2$ provides an estimate of proportion of the variance in the pooled effect size that is attributable to between-studies heterogeneity (low heterogeneity [0% to 25%], moderate [26% to 75%], and high [76% to 100%]). $p < 0.1$ for the Cochran's Q-test was considered as a significant heterogeneity among the studies [21]. Due to the significant heterogeneity between the studies, the pooled MD for each outcome was estimated using the random-effects model and the DerSimonian-Laird estimator for $\tau^2$ [22]. The analysis included calculating the mean effect size with its 95% CI and we considered a statistically significant effect when $p < 0.05$. We used a correlation value of 0.5 to calculate the variance of each study group when the studies did not report the change standard deviation (SD) or the correlations between pre-intervention and post-intervention measures. We evaluate the sensitivity of the meta-analytic results for distinct correlation values for pre-intervention and post-

intervention values (0.25, 0.5, 0.75). We conduct a sensitivity analysis to the sequential omission of every study in turn, to reflect the influence of the data from individual studies on the pooled MDs and evaluate the stability of the results. We also analyzed the impact of excluding trials that involved a small number of exercise sessions. The statistical analyses were performed using software R version 3.6.1© 2019 (R Foundation for Statistical Computing, Vienna, Austria), in RStudio environment version 1.2.1578© 2009–2019 R Studio, Inc. Narrative synthesis was carried out for outcomes where meta-analysis was not possible.

## 3. Results

### 3.1. Study Selection

The PRISMA flow diagram is outlined in Figure 1. Our search in the five databases yielded 6203 records. Through other resources, we identified three additional studies. After removing duplicates, title and abstract of the remaining 5354 results were screened. We retained 14 articles for full-text assessment and, of those, 10 did not meet eligibility criteria for reasons highlighted in Figure 1.

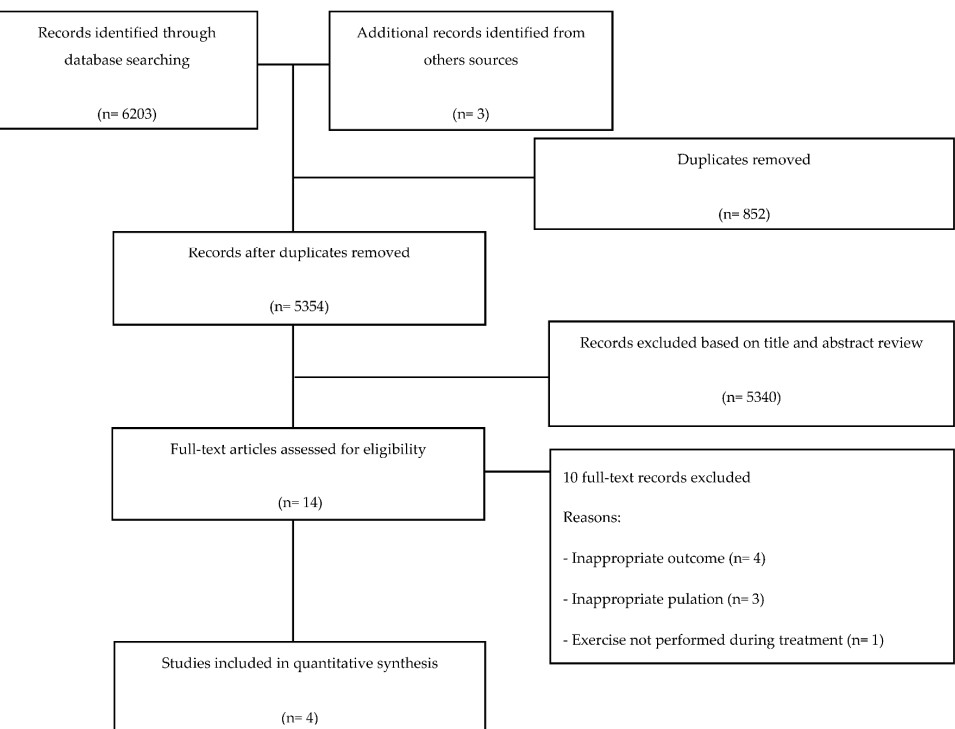

**Figure 1.** Flow of trials through the selection process.

Four unique RCTs [23–26] were included (Table 1). Furthermore, we identified ten ongoing trials [27–36] and four published study protocols [37–40] that appear to meet the predefined PICO criteria (Supplementary Material Table S2).

**Table 1.** Characteristics of the included trials.

| | Hojan et al. [23] | | | Hornsby et al. [24] | Kirkham et al. [25] | MA [26] |
|---|---|---|---|---|---|---|
| Study register | NR | | | NR | NCT02006979 | NR |
| Randomly assigned/withdrawals | $n = 68/n = 21$ | | | $n = 20/n = 1$ | $n = 27/n = 3$ | $n = 70/n = 6$ |
| Sample size | EG = 26; CG = 21 | | | EG = 9; CG = 10 | EG = 13; CG = 11 | EG = 31; CG = 33 |
| Patients' characteristics | Stage: IB-IIIA;<br>Age: EG = 54.4 ± 6.2; CG: 54.6 ± 5.2 | | | Stage: IIB-IIIC;<br>Age: EG = 46 ± 11; CG: 51 ± 6 | Stage: I-III;<br>Age: EG = 51 ± 9; CG: 50 ± 10 | Stage: NR;<br>Age: EG = 44.2 ± 5.7; CG: 43.5 ± 6.3 |
| Treatment type | TC-T | | | AC-T | AC-T | AC-T |
| Treatment setting | | | | | | |
| • Neoadjuvant | — | | | $n = 19$ (EG = 9; CG = 10) | $n = 8$ (EG = 4; CG = 4) | — |
| • Adjuvant | $n = 47$ (EG = 26; CG = 21) | | | — | $n = 16$ (EG = 9; CG = 7) | $n = 64$ (EG = 31; CG = 33) |
| • Type of training | Aerobic training | plus | Resistance training | Aerobic training | Aerobic training | Aerobic training |
| • Sequence | Linear | | Progressive | Non-linear | Linear | Linear |
| • Supervision | Supervised | | Supervised | Supervised | Supervised | Supervised |
| • Length/frequency | 9 weeks/5 TPW (TNES = 45) | | | 12 weeks/3 TPW (TNES = 36) | Single exercise session prior to each AC-T (TNES = 4) | 16 weeks/3 TPW (TNES = 48) |
| • Volume/intensity | Warm-up: 5 min/NR;<br>cool-down: 3 min/NR;<br>aerobic training: 45 min/40% to 80% HRmax; | | Sets: 1–3;<br>Reps: 8–10;<br>Load: NR; | Warm-up: NR/NR;<br>cool-down: NR/NR;<br>aerobic training: range from 15–20 min to 30–45 min/range from 60% to 100% of PO; | Warm-up: 10 min/NR;<br>cool-down: 5 min/NR;<br>aerobic training: 30-min/70% of HRR; | Warm-up: 10 min/60–70% HRmax;<br>cool-down: 10 min/60–70% HRmax;<br>aerobic training: 30 min/50–95% HRmax; |
| • Attendance rate to exercise intervention | 98.7% | | | 82% | 100% | NR |
| • Safety | No adverse events reported | | | One non-serious event (leg pain) | No adverse events | NR |
| Outcomes of interest | | | | | | |
| • Cardiac outcomes | LVEF; GLS; E/A ratio | | | LVEF | LVEF; LV longitudinal strain; E/A ratio | LVEF; E/A ratio |
| • Circulating biomarkers | NR | | | NR | NT-proBNP; cTnT | NT-proBNP; |
| Timeline assessments | -Baseline: 3 to 6 months after the start of trastuzumab<br>-Post-intervention: After 9 weeks of exercise intervention | | | -Baseline; NR<br>-Post-treatment: after 12 weeks of exercise intervention | -Baseline: 0 to 14 days prior the start of AC-T<br>-Post-treatment: 7 to 14 days after AC-T | -Baseline: NR<br>-Post-treatment: after 6 and 12 months AC-T |

Aerobic training: (duration/target intensity); cool-down: (duration/target intensity); warm-up: (duration/target intensity). Abbreviations: AC-T: anthracycline-containing therapy; CG: control group; cTnT: cardiac Troponin T; EG: exercise group; GLS: global longitudinal strain; HRmax: heart rate maximum; HRR: heart rate reserve; LV: left ventricular; LVEF: left ventricular ejection fraction; NR: not reported; PO: power output; Reps: repetitions; TC-T: trastuzumab-containing therapy; TPW: times per week; TNES: total number of exercise sessions.

### 3.2. Overview of the Included Trials

Main characteristics are described in Table 1. The four included trials involved a total of 161 women with BC. Provided cohort were middle age (mean age = 49; SD: 4.3). Three trials [24–26] were conducted during AC-T (114 patients) and one trial [23] during adjuvant trastuzumab (47 patients). All the trials implemented supervised exercise interventions. One trial [23] delivered an exercise intervention, with five sessions per week for nine weeks, combining aerobic training (AT) and resistance training (RT). The AT was performed at very light to vigorous intensity (40% to 80% of maximum heart rate) for 45 min following a linear approach and the RT involved eight different exercises, with one to five sets of eight to ten repetitions, with a progressive increase in training load. The three remaining trials adopted exercise interventions involving only AT. Kirkham et al. [25] explored the effect of a "short dose" exercise intervention which involved a single 30 min bout of aerobic exercise, at a vigorous intensity, performed 24 h prior to each AC-T cycle (i.e., maximum of four exercise sessions). The two other trials implemented an exercise intervention with three sessions per week for 12 [24] and 16 weeks [26], respectively. Hornsby et al. [24] implemented a nonlinear exercise approach with duration ranging from 15–20 min to 30–45 min and intensity ranging from 60% to 100% of power output. Lastly, Ma [26] adopted a linear exercise intervention, with 50 min at intermittent intensity ranging 50% to 95% of maximum heart rate.

### 3.3. Risk of Bias

A summary of the judgements for the risk of bias is presented in Figure 2. Selection bias has clearly been described in three trials. The remaining trial was judged to be at unclear risk for adequate random sequence generation, given the lack of clarity in methods, and to be at high risk for concealment of participant, because the allocation process was not mentioned. All trials were considered at high risk for performance bias because it was not possible to blind the participants to exercise intervention. All trials reported blinded assessment of outcomes. Two trials was judged at high risk of bias concerning incomplete data. The outcomes described in the methods section were reported in all trials. Three trials reported adherence to exercise sessions and were judged at low risk.

### 3.4. Effects of Intervention on Primary Outcome
Resting Left Ventricular Ejection Fraction

All the studies provided data concerning absolute change on LVEF. Three trials [24–26] reported data from baseline to after a short to medium-term AC-T exposure (<6 months) and one trial [23], conducted during adjuvant trastuzumab, provided data from baseline to after the exercise intervention. Pooling data across the four trials showed no significant change between the exercise group compared to control (MD: 2.07%; 95% CI: −0.17 to 4.34; $I^2$ = 38%; Figure 3A). Similar results were verified by pooling results from the three trials conducted during AC-T (MD: 1.44%; 95% CI: −1.76 to 4.65; $I^2$ = 43%; Figure 3B). The pooled data from trials that implemented interventions involving ≥36 exercise sessions (i.e., excluding data from Kirkham et al. [25]), showed a significant effect in preventing LVEF decline favoring the exercise (MD: 3.25%; 95% CI: 1.20 to 5.31; $I^2$ = 0% Figure 3C). Ma [26] was the only author that reported data on LVEF following a long-term follow-up (after 12 months of AC-T). The results demonstrated a significant attenuation in LVEF decline in the exercise group compared to control (MD: 3.80%; 95% CI: 0.49 to 7.11).

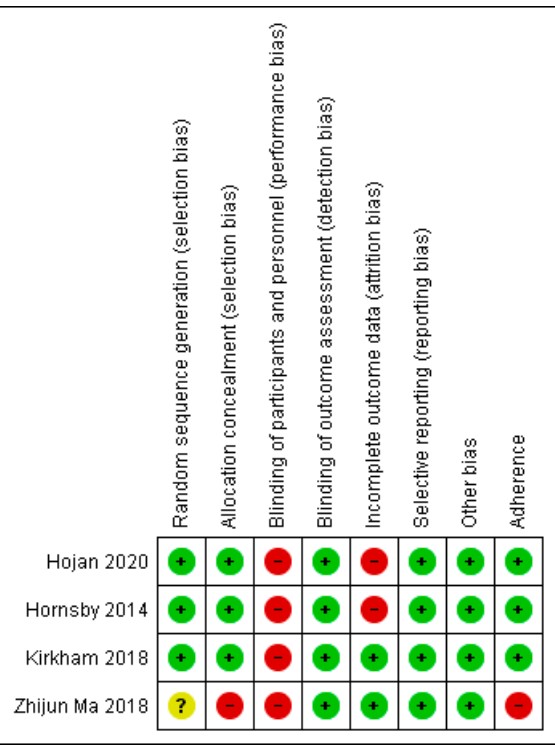

**Figure 2.** Summary of the judgements for the risk of bias of each study.

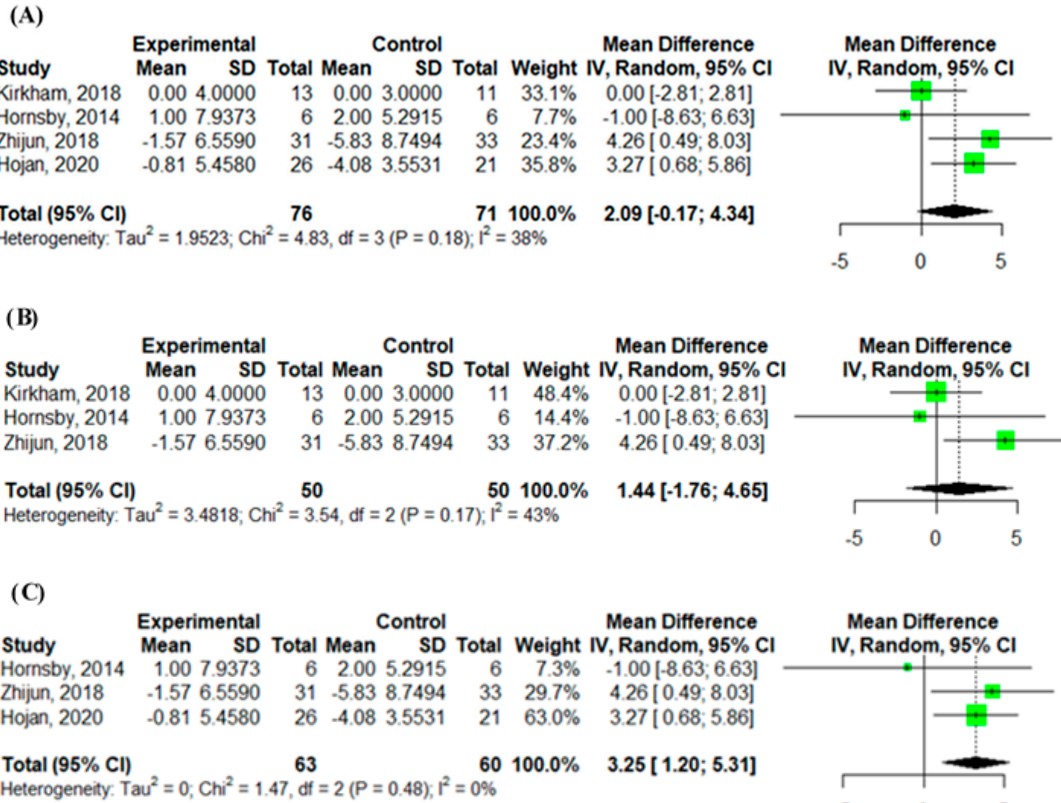

**Figure 3.** Forest plot showing the effect of exercise intervention compared to control on resting left ventricular ejection fraction: (**A**) from the included trials; (**B**) from the trials conducted during anthracycline-containing therapy; (**C**) from the trials that implemented interventions involving ≥36 exercise sessions.

### 3.5. Effects of Intervention on Secondary Outcomes

3.5.1. Resting Parameters on Strain Imaging

Kirkham et al. [25] reported data for left ventricular longitudinal strain and Hojan et al. [23] analyzed global longitudinal strain (GLS). Pooled data demonstrated no differences (MD: −0.31%; 95% CI: −2.07 to 1.46; $I^2$ = 68%; Figure 4). Kirkham et al. [25] also presented data on left ventricular twist but it was observed a null effect (MD: 0.12°; 95% CI: −0.1 to 0.35).

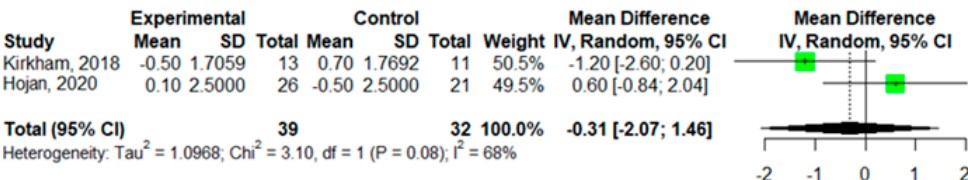

**Figure 4.** Forest plot showing the effect of exercise intervention compared to control on longitudinal strain.

3.5.2. Resting E/A Ratio

Three trials [23,25,26] reported data on E/A ratio. The pooled data revealed a null effect (MD: 0.12; 95% CI: −0.1 to 0.35; $I^2$ = 25%) (Supplementary Material, Figure S1).

3.5.3. NT-proBNP and Troponin I

Two studies [25,26] provided results concerning NT-proBNP. Pooling data showed a null effect (MD: −13.91pmo/L; 95% CI: −49.32 to 21.51; $I^2$ = 99%) (Supplementary Material, Figure S2). Only Kirkham et al. [25] reported data regarding troponin I but no significant changes were verified (MD: 2.20 pg/mL; 95% CI: −4.92 to 9.32).

## 4. Discussion

Exercise training has been suggested as a potential approach to mitigate cancer treatment-related cardiotoxicity [41,42]. There are several reviews that describe the potential protective mechanisms of exercise against cardiotoxicity, but most of these only provide narrative synthesis [43,44], data from animal studies [45,46] or are limited to cardiorespiratory fitness outcomes [47,48]. The present systematic review adds to existing literature by providing the first report of pooled data from RCTs that analyzed the impact of exercise on cardiac function outcomes and circulating biomarkers among women with BC receiving AC-T or TC-T, and a systematic compilation of ongoing trials which are researching this subject. This work demonstrated that exercise appears to have a beneficial effect to mitigating LVEF decline in women with BC receiving AC-T or TC-T, and this effect was significant for interventions with ≥36 exercise sessions. Although this new finding supports the rationale of exercise as a promising approach for preventing cardiac dysfunction, this result should be interpreted cautiously, because it was based on a limited number of RCTs which involved a small sample size, considerable methodological bias, different exercise interventions, and short-term follow-up.

Anthracycline-related cardiotoxicity is often characterized by a continuous progressive decline in LVEF, which predisposes for the development of overt heart failure [9]. Three included trials compared the effects of aerobic exercise versus usual care in LVEF among women with BC receiving AC-T [24–26]. Two of those studies reported a null effect [24,25], while Ma [26] found a significant efficacy of exercise (MD: 4.26%; 95% CI: 0.49 to 8.03). This discrepancy is likely to be explained by differences in sample size, protocol of exercise intervention, and, perhaps above all, due to the fact that MA [26] has presented data following six months of AC-T exposure, which represented a longer follow-up compared to the other two studies. More recently, in a non-randomized trial, Howden et al. [49] tested the effects of an exercise intervention (eight to twelve weeks), combining AT and RT, on cardiac function of 28 women with BC receiving AC-T. The authors found a significant

efficacy of exercise to mitigating cardiorespiratory fitness impairment ($VO_2$ peak), but the results regarding LVEF were congruent with the findings from Hornsby et al. [24] and Kirkham et al. [25] and no significant changes were verified after a short-term AC-T exposure. However, in a long-term follow-up analysis (12 months after AC-T) involving 17 participants (61%) from the initial cohort, Foulkes et al. [50] observed that compared to baseline, LVEF did not change significantly in the exercise group (MD: $-3.1$%; 95% CI: $-8.4$, to 2.1), whereas it statistically decreased in control (MD: $-6.9$%; 95% CI: $-11.1$ to $-2.0$). Moreover, two participants from the control group were diagnosed with asymptomatic cardiotoxicity. Although a possible cardioprotective effect of exercise can be hypothesized, it is plausible that these differences may also be justified by the fact that the control group has presented: older age ($58 \pm 8$ years vs. $46 \pm 9$ years; $p = 0.012$); higher body mass index ($28.1 \pm 5.5$ kg/m$^2$ vs. $24 \pm 3.3$ kg/m$^2$; $p = 0.07$) and a larger rate of patients undergoing trastuzumab (63% vs. 22%; $p = 0.15$); which are recognized factors to further compound the risk for cardiotoxicity. Furthermore, one included trial reported data on LVEF decline from a long-term follow-up (12 months after AC-T), verifying a significant attenuation in the exercise group compared to control (MD: 3.80%; 95% CI: 0.49 to 7.11) [26]. Together, these facts may hint that the effect of exercise in preventing cardiac dysfunction, assessed by LVEF, may only be detected after long-term of AC-T, which suggests that studies designed to explore this topic need to include a long-term follow-up. Moreover, further studies are needed to conclude whether the exercise training performed during AC-T provides adaptations to protect the heart in the long-term and whether this potential cardioprotective effect would extend to prevent cardiotoxicity during trastuzumab.

Regarding the setting of TC-T, we only have identified one RCT [23] and one published study protocol [38] which aimed to evaluate the effects of a high-intensity interval training (HIIT) intervention. In the unique RCT conducted during adjuvant trastuzumab, Hojan et al. [23] showed that an exercise intervention combining AT and RT was able to significantly alleviate the decline in LVEF and in aerobic capacity (assessed by the six min walk test) among the exercise group, a phenomenon that occurred in the control. Previously, in a single-arm trial, Haykowsky et al. [51] verified that AT did not prevent a significant decline in LVEF and this was not related to previous AC-T, trastuzumab dose and adherence to training sessions. The authors also reported that trastuzumab was discontinued in three patients (18%) before the post-intervention assessment. Asymptomatic decline in LVEF is the most common manifestation of cardiotoxicity during trastuzumab and can lead to premature interruption or discontinuation of therapy [5]. Although often considered reversible after treatment discontinuation (Type II cardiotoxicity) [5], Yu et al. [52] verified that patients with versus without history of trastuzumab-related cardiotoxicity presented lower mean values of resting LVEF ($56.9 \pm 5.2$% vs. $62.4 \pm 4$%; $p < 0.001$), GLS ($-17.8 \pm 2.2$% vs. $-19.8 \pm 2.2$%; $p < 0.001$) and $VO_2$ peak ($22.9 \pm 4.4$ mL/kg/min vs. $27 \pm 5.3$ mL/kg/min; $p < 0.001$) after a mean seven-year follow-up. These novel findings suggest that the development of cardiotoxicity during trastuzumab may lead to a persistent long-term impairment on cardiac function, which extends beyond the heart, affecting the cardiorespiratory capacity, emphasizing the need to implement holistic strategies for support cardiovascular health. Previous meta-analysis [15,47,48] have highlighted the effectiveness of exercise training to prevent or mitigate the decline of cardiorespiratory capacity.

Although widely used for monitoring cardiac function in cardio-oncology setting, LVEF has important limitations [10,11], thus, an integrated approach which includes the assessment of other echocardiographic outcomes (parameters of diastolic function and strain imaging), and biomarkers has been recommended. Three [23,25,26] and two [25,26] included trials reported, respectively, data on diastolic function (measured by E/A ratio), and NT-proBNP. For both outcomes, pooled analysis did not reveal significant changes. Regarding strain imaging outcomes, two studies analyzed longitudinal strain [23,26] and GLS [23], but no changes were verified. In contrast, findings from a non-randomized study showed that exercise training, combining AT and RT, during AC-T, mitigated the

GLS decline [53]. This is a relevant finding, that needs to be confirmed, as GLS has been suggested to early predicts subclinical cardiac dysfunction [10,11].

The limited number of trials in this review did not allow us to analyze the role of different training types and intensity on study outcomes. In light of the literature of the cardiac rehabilitation field, continuous aerobic exercise, at moderate intensity, has been associated to improve LVEF in heart failure patients, but that benefit was not evident when RT was performed alone or combined with AT [54–56]. However, Hojan et al. [23] demonstrated that AT combined to RT had a favorable effect in preventing LVEF decline among women with BC receiving adjuvant trastuzmab. Furthermore, Tucker et al. [56] highlight that the efficacy of moderate intensity continuous aerobic exercise on LVEF and $VO_2$ peak are greatest in long-term interventions (length $\geq$ six months). Such results emphasize the importance of implementing interventions that last longer than the AC-T regime; which range from eight (dose-dense protocol) to twelve weeks (every three weeks protocol). Therefore, additional research is needed to identify a safe and optimal dose of FITT (frequency, intensity, time, type) prescription to target this issue. We identified ten ongoing trials [27,36] and four published study protocols [37–40] which are exploring this topic and their results might help to address some research gaps in a near future.

Despite of its originality and adherence to a published protocol and to PRISMA guidelines, when interpreting the findings of this review, one should bear in mind a few limitations. The main limitation of this review was the scarce of RCTs included, which reflects the lack of clinical trial evidence in the literature addressing this topic. It is possible that the strict eligibility criteria applied (only RCTs) and the fact that this is relatively recent research topic, have contributed to this limited selection. Moreover, included trials presented methodological differences and are constrained by its methodological quality (i.e., underpowered studies, high risk of bias, and short-term follow-up data to evaluate clinical events). Second, there was observed a considerable heterogeneity in the components of the exercise training interventions (i.e., F = frequency, I = intensity, T = time, and T = type) across the included studies. Third, we only identified one trial conducted during TC-T. Fourth, outcomes of diastolic function and strain imaging were not consistently assessed among the trials. Fourth, trials had a relatively short-term follow-up, and a longer-term follow-up may be needed to understand the real impact of exercise on cardiac function of cancer patients treated with cardiotoxic therapies. Finally, cardiorespiratory fitness is well established as a strong predictor of cardiovascular and all-cause mortality among healthy individuals and in several patient populations, including cancer survivors, and thus, it should be addressed in future updates of this review.

Therefore, future RCTs with the following focus is widely encouraged: (a) trials designed for a long-term intervention and follow-up; (b) enroll elderly patients; (c) explore the effects of different training modalities (i.e., AT combining RT; HIIT combining or not with RT) and characteristics (i.e., linear vs. non-linear approach); (d) implement exercise interventions during AC-T and which may also be extensible for TC-T; (e) consider the analysis of promising parameters for the prediction of subclinical cardiotoxicity (i.e., as strain imaging; diastolic function outcomes and circulating biomarkers).

## 5. Conclusions

Exercise appears to exert a beneficial effect for mitigating LVEF decline in women with BC receiving AC-T or TC-T, and this effect was significant for interventions with $\geq$36 exercise sessions. Although this finding supports the rationale of exercise as a promising approach for preventing cancer therapy-related cardiotoxicity, this result should be interpreted cautiously, because it was based on a limited number of RCTs. Thus, more trials are warranted to validate the efficacy of exercise in preventing cancer therapy-related cardiotoxicity and to analyze its effects on subclinical cardiotoxicity outcomes (strain imaging, diastolic function and circulating biomarkers). We identified ten ongoing trials researching this issue and their future results can be used to upgrade this review.

**Supplementary Materials:** The following are available online at https://www.mdpi.com/article/10
.3390/app11188336/s1, Figure S1: Forest plot showing the effect of exercise intervention compared
to control on E/A ratio, Figure S2: Forest plot showing the effect of exercise intervention compared
to control on NT-proBNP, Table S1: search strategy used in PubMed, Table S2: characteristics of the
identified ongoing trials.

**Author Contributions:** The idea of the above-mentioned study was conceived and designed by P.A.
and D.E. All authors contributed for the development of the study methodology. The search and
trials selection were performed by P.A. and D.E., A.J. and A.A. (Anabela Amarelo) assessed the risk
of bias. Data extraction was conducted by P.A., C.N., A.A. (Anabela Amarelo) and H.C. data analysis
was performed by C.N. and V.A. The data synthesis was conducted by P.A, H.C., A.A. (Alberto
Alves), J.F.-M. The first draft was written by P.A., D.E., A.A. (Alberto Alves) and A.J. All authors
reviewed, wrote and approved the final version. All authors have read and agreed to the published
version of the manuscript.

**Funding:** Pedro Antunes was supported by an individual doctoral grant from Fundação para a
Ciência e a Tecnologia (SFRH/BD/143226/2019). This work is financed by National Funds through
Fundação para a Ciência e a Tecnologia under the project UID04045/2020.

**Institutional Review Board Statement:** Not applicable.

**Informed Consent Statement:** Not applicable.

**Data Availability Statement:** Not applicable.

**Acknowledgments:** This study was part of a Ph.D. project conducted at Sports Science Department
of University of Beira Interior. The authors are grateful to the Center for Research and Develop-
ment in Mathematics and Applications (CIDMA) through the Portuguese Foundation for Science
and Technology (FCT-Fundação para a Ciência e a Tecnologia), references UIDB/04106/2020 and
UIDP/04106/2020.

**Conflicts of Interest:** The authors declare no conflict of interest.

## Abbreviations

AC-T: anthracycline-containing therapy; AT: aerobic training; BC: breast cancer; FITT: frequency,
intensity, time, type; HIIT: high-intensity interval training; GLS: global longitudinal strain; LVEF: left
ventricular ejection fraction; MD: mean difference; NT-proBNP: n-terminal pro-b-type natriuretic
peptide; PICO: population-intervention-control outcome; RCT: randomized controlled trial; RT:
resistance training; SD: standard deviation; TC-T: trastuzumab-containing therapy.

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
