# Peer review of "Effects of Exercise on Cardiac Function Outcomes in Women Receiving Anthracycline or Trastuzumab Treatment for Breast Cancer: A Systematic Review and Meta-Analysis"

_applsci, doi:10.3390/app11188336_

Round 1
Reviewer 1 Report
This is a systematic review of exercise intervention as a protection against loss of cardiac function during chemotherapy.
Introduction - OK
Methods:
"no difference" or null hypothesis was the finding in most of the analyses, apart from the ">36 sessions" of one study
The Forest plot adequately demonstrates the amount of statistical confidence to have in a "no difference" finding .
Discussion:
the study is hypothesis generating:
- why were few differences found?
- would a bigger sample have shown a difference in any of the studies
- is there a biological hypothesis behind the "no difference"
- the one positive finding - how would this be validated?
Conclusion
appropriate
Author Response
Response to Reviewer 1 Comments
- We sincerely thank the reviewer for the analysis of our manuscript and for the comments that helped the authors to improve the quality of the new version of the work.
Point 1: Why were few differences found? would a bigger sample have shown a difference in any of the studies?
R: Thank you for your questions. We believe that the few differences presented are justified by the limited number of studies involved (n=4); by the relative short follow-up; by methodological constraints; and by the differences in the intervention applied in each study. We chose to add to Discussion 4. the following description of the main limitations that help in interpreting the results: “line 388: Despite of its originality and adherence to a published protocol and to PRISMA guidelines, when interpreting the findings of this review, one should bear in mind a few limitations. The main limitation of this review was the scarce of RCTs included, which reflects the lack of clinical trial evidence in the literature addressing this topic. It is possible that the strict eligibility criteria applied (only RCTs) and the fact that this is relatively recent research topic, have contributed to this limited selection. Moreover, included trials presented methodological differences and are constrained by its methodological quality (i.e., underpowered studies, high risk of bias, and short-term follow-up data to evaluate clinical events). Second, there was observed a considerable heterogeneity in the components of the exercise training interventions (i.e., F = frequency, I = intensity, T = time, and T = type) across the included studies. Third, we only identified 1 trial conducted during TC-T. Fourth, outcomes of diastolic function and strain imaging were not consistently assessed among the trials. Finally, trials had a relatively short-term follow-up, and a longer-term follow-up may be needed to understand the real impact of exercise on cardiac function of cancer patients treated with cardiotoxic therapies.
Point 2: is there a biological hypothesis behind the "no difference".
The heterogeneity of the exercise training interventions across the included studies (Length; Frequency; Duration and Intensity) does not allow us to draw conclusions. The intensity of aerobic training is described as a fundamental variable to mitigate the decline in cardiac function, and it is suggested that the main benefits come from vigorous intensity [24]. Although it has been established that moderate-intensity aerobic training is well tolerated among women with breast cancer undergoing chemotherapy, vigorous-intensity interventions may not be tolerable for a significant portion of this population (Hornsby et al. [24] reported that 23% (68/296) of sessions required a reduction in exercise duration and/or intensity).
Point 3: the one positive finding - how would this be validated?
To validate the promising result presented in this study, which goes beyond the already known benefits in cardiorespiratory fitness, future RCTs should consider some methodological components that have been added to the discussion: line 402 “Therefore, future RCTs with the following focus is widely encouraged: (a) trials designed for a long-term intervention and follow-up; (b) enrol elderly patients; (c) explore the effects of different training modalities (i.e., AT combining RT; HIIT combining or not with RT) and characteristics (i.e., linear vs non-linear approach); (d) implement exercise interventions during AC-T and which may also be extensible for TC-T; (e) consider the analysis of promising parameters for the prediction of subclinical cardiotoxicity (i.e., as strain imaging; diastolic function outcomes and circulating biomarkers).”.

Reviewer 2 Report
The review is very interesting , complete and well conducted .
The first observation is that it related to a selection of supervised exercise manuscripts and therefore I think it is mode adequate to clarify this choice also in the title .
The parallel positive effect of unsupervised exercise could be written in the discussion or in the introduction part .
There are some evidence of the positive effects of the unsupervised exercise , with a longer adherence to individual programs . A specific experience reports some evidence especially in strain and ain diastolic function distinguishing the two diverse physical activity : sports and moderate unsupervised physical exercise .
Dragon Boat training exerts a positive effect on myocardial function in breast cancer survivors. Laura Stefani, Giorgio Galanti, Valentina Di Tante, Riggs J. Klika and Nicola Maffulli. Phys Sportsmed, 2015; DOI: 10.1080/00913847.2015.1037711
In the introduction , the authors should also specify the definition of Cardiotoxicity in order to better define the difficulties to identify the initial phase of an eventual heart’s failure especially by the standard echo parameters. Some aspect of the importance of the deformation parameters for the clinical practice have been partially approached.
- Regarding the inclusion criteria the authors say ….To be eligible, the exercise intervention had to be performed during AC-T or TC-T. Trials were not considered if Yoga, Tai Chi Chuan, Qigong or Pilates was defined as exercise modality
- Consider that this is one of the most common kind of exercise indicated in breast cancer during chemotherapy . It is well known as these physical activities do not increase significantly the LVEF , however there were within the moderate intensity of exercise, and therefore specific for cancer patients. The authors should specify the range of the intensity of the exercise prescribed and in any case clarify the exclusion of Yoga, Tai Chi Chuan, Qigong or Pilate from the present study.
Regarding the secondary outcomes the authors report as … “also presented data on left ventricular twist but it was observed a null effect”
This sentence needs to be interpreted and justified as consequence of the fact it is known that the circumferential strain contributes to the maximal myocardial performance, especially in case of regular training
The authors talk about … significantly alleviate the decline in LVEF in the exercise group…. It could be important to cite also the “tolerance to the exercise” , often reported following the CR 10 or BORG scale . This parameter , as the VO2, mas seems not cited in any part of the review. It could be important to give more strength to the paper, to talk about this aspect in the discussion as a limit of the investigation and as a future perspective.
Author Response
Response to Reviewer 2 Comments
Point 1: The review is very interesting, complete and well conducted.
R: We wish to sincerely thank the reviewer for the analysis of our manuscript and for the comments that helped the authors to improve the quality of the new version of the work.
Point 2: The first observation is that it related to a selection of supervised exercise manuscripts and therefore I think it is mode adequate to clarify this choice also in the title. The parallel positive effect of unsupervised exercise could be written in the discussion or in the introduction part. There are some evidence of the positive effects of the unsupervised exercise, with a longer adherence to individual programs. A specific experience reports some evidence especially in strain and ain diastolic function distinguishing the two diverse physical activity: sports and moderate unsupervised physical exercise. Dragon Boat training exerts a positive effect on myocardial function in breast cancer survivors. Laura Stefani, Giorgio Galanti, Valentina Di Tante, Riggs J. Klika and Nicola Maffulli. Phys Sportsmed, 2015; DOI: 10.1080/00913847.2015.1037711
R: We would like to clarify that the mode of implementation of the exercise training interventions (i.e., supervised, home-based or combination) was not considered as an eligibility criterion, although the 4 articles included in this meta-analysis implemented supervised interventions. We chose not to add to the manuscript title: a systematic review and meta-analysis of randomized controlled trials as it seemed to us that it would be too extensive.
Point 3: In the introduction, the authors should also specify the definition of Cardiotoxicity in order to better define the difficulties to identify the initial phase of an eventual heart’s failure especially by the standard echo parameters. Some aspect of the importance of the deformation parameters for the clinical practice have been partially approached.
R: We agree with the reviewer, and we added the definition of cardiotoxicity “Cardiotoxicity is defined by a decrease in left ventricular ejection fraction (LVEF) of > 10% to a value below the lower limit of normal”.
Point 4: Regarding the inclusion criteria the authors say … To be eligible, the exercise intervention had to be performed during AC-T or TC-T. Trials were not considered if Yoga, Tai Chi Chuan, Qigong or Pilates was defined as exercise modality. Consider that this is one of the most common kind of exercise indicated in breast cancer during chemotherapy. It is well known as these physical activities do not increase significantly the LVEF, however there were within the moderate intensity of exercise, and therefore specific for cancer patients. The authors should specify the range of the intensity of the exercise prescribed and in any case clarify the exclusion of Yoga, Tai Chi Chuan, Qigong or Pilate from the present study.
R: Thank you for your comment. There is a wide spectrum of scientific evidence pointing the benefits of these type of modalities in several physiological and psychological outcomes among breast cancer survivors (Myeong Soo Lee. Qigong for cancer treatment: a systematic review of controlled clinical trials. Acta Oncol . 2007;46(6):717-22. doi: 10.1080/02841860701261584.; Karen M. Mustian. Tai Chi Chuan for Breast Cancer Survivors. Med Sport Sci. 2008; 52: 209–217. doi: 10.1159/000134301) and in patients with heart failure (Lei Pan et al. Effects of Tai Chi training on exercise capacity and quality of life in patients with chronic heart failure: a meta-analysis. Eur J Heart Fail . 2013. doi: 10.1093/eurjhf/hfs170). Benefits on cardiac function in patients with heart failure have also been reported (Xiaomeng Ren et al. The Effects of Tai Chi Training in Patients with Heart Failure: A Systematic Review and Meta-Analysis. 2017 doi: 10.3389/fphys.2017.00989; Jinke Huang eta al. The Effects of Tai Chi Exercise Among Adults With Chronic Heart Failure: An Overview of Systematic Review and Meta-Analysis. Front. Cardiovasc. Med, 2021 | https://doi.org/10.3389/fcvm.2021.589267). However, in this meta-analysis we only considered interventions that involved aerobic training and resistance training, or a combination of both. The reasons that justify this decision were: the modalities of Yoga, Tai Chi Chuan, Qigong or Pilate can be implemented with different types of style, making this heterogeneity a limitation; aerobic training and resistance training are the types of training most implemented and investigated in this population, and in particular, aerobic training is the type of training that is hypothesized to mitigate cancer-related cardiotoxicity, in this case, the most promising results are expected, and therefore a detailed description of the intervention with definition of FITT parameters will be indicated for readers.
Point 5: The authors talk about … significantly alleviate the decline in LVEF in the exercise group…. It could be important to cite also the “tolerance to the exercise”, often reported following the CR 10 or BORG scale. This parameter, as the VO2, seems not cited in any part of the review. It could be important to give more strength to the paper, to talk about this aspect in the discussion as a limit of the investigation and as a future perspective.
R: Thank you for your comment. Cardiorespiratory fitness is well established as a strong predictor of cardiovascular and all-cause mortality among healthy individuals and in several patient population, including cancer survivors. In fact, in addition to mitigating the decline in LVEF, Hojan et al. [23] found a similar effect on aerobic capacity measured by the 6MWT. We changed our sentence to: “Hojan et al. [23] showed that an exercise intervention combining AT and RT was able to significantly alleviate the decline in LVEF and in aerobic capacity (assessed by the 6-minute walk test)”. In the discussion, line 336-337, we had already highlighted the significant effect of exercise in attenuating the decline of VO2peak during anthracycline-containing chemotherapy. In line 368-375, we present data pointing that cardiotoxicity extends beyond the heart and affecting significantly cardiopulmonary function which may contribute to increased risk of late-occurring cardiovascular disease in survivors of ERBB2-positive breast cancer. However, given that cardiorespiratory capacity was not defined as a study outcome, we do not broadly address this subject. In line 375-376, “Previous meta-analysis [15,47, 48] have highlighted the effectiveness of exercise training to prevent or mitigate the decline of cardiorespiratory capacity”. were added to 4. Discussion to guide authors to previous meta-analyses that address this point.
However, we agree with the reviewer that cardiorespiratory capacity is an important parameter that should be included as a study outcome in a future update of this review (we added line: 414-416: “Finally, cardiorespiratory fitness is well established as a strong predictor of cardiovascular and all-cause mortality among healthy individuals and in several patient population, including cancer survivors, and should be adressed in future updates of this review.”).
